# The Effect of Aging Treatment on the Corrosion Behavior of 17-4PH Stainless Steel

**DOI:** 10.3390/ma18081823

**Published:** 2025-04-16

**Authors:** Chengshuang Zhou, Yin Lv, Lin Zhang

**Affiliations:** College of Materials Science and Engineering, Zhejiang University of Technology, Hangzhou 310014, China; 17857687416@163.com

**Keywords:** 17-4PH stainless steel, Cu-rich precipitates, dislocations, corrosion resistance, corrosion morphology

## Abstract

This study systematically investigated the influence of aging temperature variations on the evolution of Cu-rich precipitates and dislocation distribution characteristics in 17-4PH stainless steel through comprehensive electrochemical testing and microstructural characterization. The mechanism by which microstructural features govern electrochemical corrosion behavior was elucidated. Experimental results demonstrated that within the aging temperature range of 480–620 °C, matrix dislocations consistently maintained non-uniform distribution characteristics, though their regional heterogeneity exhibited a decreasing trend with increasing temperature. The precipitation behavior of copper followed an evolutionary sequence: transitioning from dispersed copper precipitates to finely distributed Cu-rich precipitates with high numerical density, ultimately progressing to coarsening and agglomeration. The corrosion resistance of the material initially improved before subsequent degradation, accompanied by a morphological transition of surface corrosion features from characteristic elongated striations to elliptical patterns. Samples aged at 580 °C for 4 h exhibited optimal corrosion resistance. Mechanistic analysis revealed that reduced dislocation density heterogeneity effectively minimized electrochemical potential differences between micro-regions, while elemental segregation induced by Cu-rich precipitates coarsening intensified local electrochemical inhomogeneity. These two mechanisms cooperatively regulated the overall corrosion resistance evolution of the material.

## 1. Introduction

Stainless steel technology has played a significant role in addressing corrosive environments across various industrial sectors [1,2]. The development of precipitation-hardened stainless steels aims to provide high strength and toughness while maintaining the superior corrosion resistance inherent to stainless steels [3]. As a representative martensitic stainless steel, 17-4PH (UNS S17400) has gained extensive applications in aerospace engine components [4,5], marine engineering fasteners [6], and petrochemical/nuclear power systems [7,8], owing to its favorable corrosion resistance combined with high strength and toughness achieved through precipitation formation during aging treatment [9,10]. Previous investigations have primarily focused on composition optimization [2], surface treatments, and corrosion mechanisms of martensitic stainless steels [11,12], particularly their performance in aggressive environments such as acidic media [13] and chloride-containing solutions [14]. In the harsh corrosive environment of the offshore oil industry, long-term exposure to high salt spray, high humidity, and chlorine-rich seawater media lead to the initiation of pitting pits and the intensification of the intergranular corrosion tendency of materials, which poses a continuous challenge to the corrosion resistance of materials [15,16]. Therefore, it is particularly important to optimize the corrosion resistance of 17-4PH stainless steel. Research has established that the corrosion resistance of martensitic stainless steels is intrinsically linked to their microstructural characteristics [17,18,19]. Studies have shown that [20] in the range of 940–1140 °C, 17-4PH stainless steel can obtain higher strength and also show good plasticity after water quenching at 1040 °C. In order to obtain higher comprehensive properties, 17-4PH stainless steel is usually further aged. It is reported that 17-4PH stainless steel aged at 480 °C has the highest strength, which is also known as the peak aging temperature [21,22,23]. It is worth mentioning that previous studies [24] have shown that 17-4PH aged at 480 °C for 1 h has only slightly segregated Cu-rich precipitates rather than a large amount of agglomerated precipitation. When the aging temperature continues to rise to more than 550 °C, a large number of Cu-rich precipitates are deposited, and large precipitates lead to a decrease in hardness and strength, but an increase in plasticity and toughness [22]. However, when the aging temperature further increases to more than 600 °C, the strength decreases and the plasticity cannot be increased at the same time, and the aging treatment at this time is called over-aging [9,23,25]. At the same time, as the aging temperature increases, the dislocation of 17-4PH will decrease correspondingly. Regarding the relationship between dislocation density and the corrosion resistance of materials, studies [26,27] have shown that high-density dislocation will lead to the increase of corrosion active sites in materials. However, when the dislocation is uniformly distributed, it can be used as the nucleation site of the passivation film, providing power for the formation of the passivation film, and thus contributing to the uniform formation and thickening of the passivation film, which is thus conducive to the corrosion resistance of the material [28,29]. Some studies [30,31] have also shown that excessive local dislocation density difference lead to excessive electrochemical potential difference in different regions, which promotes the dissolution of metals and forms voidage, which will lead to the deterioration of the corrosion resistance of materials. Although a large number of studies have discussed the influence of alloying elements and corrosion environment on the corrosion resistance of 17-4PH [32,33], systematic studies on the influence of aging treatment, especially the temperature range from peak aging to over-aging, on the microstructure of martensitic stainless steel and its mechanism of action on electrochemical corrosion performance have been conducted. In particular, the synergic change of copper-rich precipitated phase and dislocation density in the 17-4PH aging process has little influence on its corrosion resistance. Consequently, this study focuses on elucidating the effects of aging treatment on the microstructural evolution of 17-4PH stainless steel and further analyzes its implications for corrosion resistance, aiming to provide theoretical foundations for enhancing the material’s performance in corrosive service environments.

## 2. Experiments

### 2.1. Materials and Samples

The present study employed commercially available 17-4PH (UNS S17400) martensitic precipitation-hardened stainless steel, with its chemical composition provided by the manufacturer as shown in Table 1.

The stainless steel samples underwent solution treatment at 1040 °C for 1 h followed by water quenching. Subsequently, the quenched steel was subjected to aging treatments at 480 °C for 1 h, and at 520 °C, 550 °C, 580 °C, 595 °C, and 620 °C for 4 h, respectively. All of the above heat treatments were performed in a vacuum environment, and the heat treatment equipment was a VHB-335H vacuum brazing furnace developed by Shenyang Jiayu Vacuum Technology Co., Ltd., (Shenyang, China). These treatments produced Cu-rich precipitates with distinct dimensions, population densities, and structural configurations, along with varying dislocation densities. The resulting samples were designated as “A480”, “B520”, “C550”, “D580”, “E595”, and “F620” corresponding to their respective aging parameters. A schematic representation of the heat treatment protocol is presented in Figure 1.

The electrochemical test samples were prepared through the following procedure: First, rectangular coupons (10 mm × 10 mm × 5 mm) were sectioned from 17-4PH stainless steel using a wire-cutting machine. Electrical leads were then welded to the samples, which were subsequently encapsulated in plastic tubes using epoxy resin to expose only a 1 cm^2^ rectangular working area. The samples were mechanically ground with 600–2000 grit SiC abrasive paper, followed by sequential polishing using 2.5, 1.0, and 0.5 μm diamond grinding pastes. Prior to electrochemical testing, the samples were ultrasonically cleaned with anhydrous ethanol and deionized water, then air-dried.

### 2.2. Electrochemical Measurements

Electrochemical tests were performed using a CHI 760E workstation (Shanghai Chenhua Instrument Co., Ltd., Shanghai, China) in a conventional three-electrode cell configuration. A platinum sheet served as the counter electrode, the sample as the working electrode, and a saturated calomel electrode (SCE) as the reference electrode. Potentiodynamic polarization tests were conducted to assess the electrochemical corrosion behavior of 17-4PH surfaces. The electrolyte consisted of 3.5 wt% NaCl solution, a standardized medium for simulating marine corrosion environments. All tests were maintained at 25 °C. Prior to measurements, samples were cathodically polarized at −1.0 V_SCE_ for 5 min to remove surface passivation films, followed by a 3600 s open-circuit potential (OCP) measurement to stabilize the electrochemical interface. Potentiodynamic polarization scans were executed over a potential range of −0.8 V to +0.8 V at a sweep rate of 0.5 mV·s^−1^. Each electrochemical test was repeated at least three times to ensure reproducibility.

### 2.3. Microstructural Characterization

Light microscopy (LM) was employed to examine the metallographic structure of 17-4PH stainless steel after quenching and aging at various temperatures. A mixed etchant solution consisting of 5 g CuCl_2_, 40 mL HCl, 25 mL C_2_H_5_OH, and 30 mL deionized water was utilized for microstructural revelation, and the metallographic erosion lasted for 15 s at room temperature. The prior austenite grain size of each sample was measured by ImageJ software (https://imagej.net/ij/, accessed on 29 July 2024). Scanning electron microscopy (SEM; FEI NovaNano equipped with energy-dispersive spectroscopy EDS, operating at 15 kV) and secondary electron (SE) imaging were applied to analyze pitting morphologies and corrosion products, with qualitative characterization of corrosion byproducts. Grain size and matrix lath variation of individual samples were studied by Electron Backscatter Diffraction (EBSD). The samples were mechanically polished and then electrolytically polished with an electrolytic solution of a mixture of perchloric acid, glycerol, and alcohol (2:1:17 by volume) at 20 V and at room temperature for 30–35 s. The model of the equipment was Zeiss-Geimi 350 (Carl Zeiss AG, Oberkochen, Germany). To investigate microstructural evolution under varying aging temperatures, thin disk samples (3 mm in diameter) were mechanically ground to 50 μm thickness, followed by twin-jet electropolishing in an electrolyte containing 10% perchloric acid and 90% ethanol at −25 °C under a 35 mA current. High-angle annular dark-field scanning transmission electron microscopy (HAADF-STEM) and high-resolution transmission electron microscopy (HRTEM) analyses were performed using a Thermo Fisher Themis Z TEM operated at 300 kV and a Talos F200X TEM (Thermo Fisher Scientific, Waltham, MA, USA) operated at 200 kV, respectively. The copper-rich phase size was calculated by ImageJ software.

## 3. Results

### 3.1. Effect of Aging Treatment on Microstructural Evolution

Light microscopy (LM) images capturing the microstructural evolution of 17-4PH stainless steel under different aging treatments are presented in Figure 2. The metallographic analysis reveals no significant variation in prior austenite grain size across samples aged within the 480–620 °C temperature range. Prior austenite grains predominantly measure 18–20 μm in diameter, with minor fractions exhibiting larger dimensions (~30 μm) and smaller grains (~10 μm). The microstructure of the A480 sample displays the highest density of martensitic laths, with progressive interfacial blurring observed as aging temperature increases. Numerous fine precipitates, indiscernible in optical micrographs due to resolution limitations, are randomly distributed across all samples. The morphological evolution of these precipitates with aging temperature remains challenging to resolve via conventional LM techniques.

The GB+IPF results of the steel after treatment with different aging temperatures are shown in Figure 3. According to the results of EBSD analysis (area-weighted method), the grain sizes of A480, B520, C550, D580, E595, and F620 are 7.6, 8.4, 8.0, 8.0, 9.4, and 7.6 μm, respectively, and the overall grain size fluctuates between 7.6 and 9.6 μm. The IPF indicates that the number of laths of 17-4PH stainless steels after different aging treatments does not change significantly, and it is found by measurement that the prior austenite grain size is mostly around 20 μm.

Transmission electron microscopy (TEM) images of four representative samples selected based on aging temperatures are presented in Figure 4: (a) A480, (b) B520, (c) D580, and (d) F620. TEM analysis qualitatively reveals a progressive reduction in dislocation density as aging temperature increases from 480 °C to 620 °C. The A480 sample exhibits a high dislocation density throughout the matrix, while B520 and D580 samples show reduced dislocation densities compared to A480. Notably, the F620 sample aged at 620 °C demonstrates a significant decline in dislocation density. These observations align with optical microscopy findings, where martensitic lath boundaries gradually lose interfacial definition with increasing aging temperature. The TEM results quantitatively corroborate the microstructural evolution associated with thermal aging conditions.

The precipitation and distribution of nano-sized Cu-rich precipitates across representative samples—(a) A480, (b) B520, (c) D580, and (d) F620—are illustrated in Figure 5. TEM micrographs reveal no discernible segregation of nano-sized copper-rich precipitates in the A480 sample, though studies [24] have shown incipient copper clustering within the matrix. Elevated aging temperatures induce distinct precipitation behavior: B520 exhibits uniformly distributed fine Cu-rich precipitates (indicated by yellow arrows), while D580 demonstrates coarsened intragranular precipitates with reduced population density compared to B520. Previous studies [25] suggest that aging above 550 °C promotes substantial precipitation of Cu-rich precipitates, where larger precipitates reduce hardness and strength while enhancing ductility and toughness. This coarsening process is attributed to the coalescence of finer precursor clusters [34,35]. Samples aged above 600 °C (e.g., F620) enter an over-aging regime [9,25], characterized by marginally increased precipitate size and reduced population density relative to D580, without corresponding improvements in ductility [9,23,25]. Comparative analysis between D580 and F620 reveals no significant redistribution of Cu-rich precipitates, suggesting saturation of coarsening mechanisms at elevated aging temperatures.

The size distribution histogram of copper-rich precipitates obtained using ImageJ software is shown in Figure 6 (the 70 copper-rich precipitates of each sample were counted to make the size distribution map of the copper-rich phase of B520, D580, and F620 samples), with (a), (b), and (c) corresponding to samples B520, D580, and F620, respectively. From the distribution histograms, it is visually apparent that the Cu-rich precipitates in sample B520 are primarily concentrated in the 5–8 nm range, while those in sample D580 are mainly clustered in the 10–16 nm range. In contrast, the Cu-rich precipitates in sample F620 are distributed within the 10–20 nm range. Additionally, as the aging temperature increases, the Cu-rich precipitates coarsen and agglomerate, with smaller precipitates gradually disappearing and the size distribution shifting toward larger dimensions.

### 3.2. Electrochemical Test Result

The potentiodynamic polarization curves of each sample in a 3.5 wt.% NaCl solution at 25 °C are displayed in Figure 7. The electrochemical parameters obtained from the dynamic polarization curves are presented in Table 2. The electrochemical results indicate that for 17-4PH, the corrosion current initially decreases and then increases as the aging temperature rises, with the D580 sample exhibiting the lowest self-corrosion current and the A480 sample showing the highest self-corrosion current. The corrosion potential generally exhibits a decreasing trend as the aging temperature increases. As seen in Figure 6, the pitting potential of each sample is quite similar (except for the B520 sample), but overall, the D580 sample has the highest pitting potential. The combined parameters from the dynamic polarization curves suggest that the D580 sample has superior corrosion resistance compared to the other samples. It is noteworthy that when pitting occurs, the current density increases dramatically. Even the A480 sample, which has the poorest corrosion resistance, sees its current density increase by two orders of magnitude, while the other five samples experience an increase of over four orders of magnitude.

The Nyquist plots and corresponding fitting circuit diagrams for 17-4PH subjected to different aging treatments are presented in Figure 8. The fitting circuit diagrams were obtained using ZView 2 software, where R1 represents the solution resistance, and R2 and R3 represent the corrosion resistances. The Nyquist plots indicate that, as the aging temperature increases, the diameter of the plots generally first increases and then decreases. The D580 sample exhibits the largest diameter, while the A480 sample has the smallest diameter. The diameters of the F620 and E595 samples are similar, with the former slightly larger than the latter. Clearly, the impedance results suggest that the D580 sample has the best corrosion resistance, while the A480 sample has the poorest corrosion resistance.

The bode plots of 17-4PH for different aging treatments are shown in Figure 9. From Figure 9a, it can be seen that the D580 sample has a large phase angle in a wide frequency range, while the A480 sample has a large phase angle in the narrowest frequency range, and other samples have a similar frequency range with a large phase angle, which indicates that the D580 sample has the best passivation behavior, while the A480 sample has the worst passivation behavior. The passivation behavior of other samples is similar. At the same time, samples with a larger value of |Z| in the low-frequency region of the bode figure indicate better corrosion resistance and higher chemical stability. It can be seen from Figure 9b that the |Z| value of sample D580 at 0.01 Hz is the largest, while that of sample A480 is the smallest, and other samples are similar. In summary, the EIS results showed that the D580 sample had the best corrosion resistance and the A480 sample had the worst corrosion resistance.

### 3.3. Surface Morphology of the Samples After Electrochemical Testing

The macroscopic surface morphology of 17-4PH after electrochemical testing in a 3.5 wt% NaCl solution, following different aging treatments, is shown in Figure 10. It can be seen from the corrosion morphology of each sample that there are two types of corrosion pits. The corrosion pits of A480 and B520 samples are elongated, different from the oval or circular corrosion pits of other samples. Therefore, the same type of A480 and B520 samples were compared, and four samples (C550, D580, E595, and F620) were compared separately. The corrosion pit of the A480 sample is significantly larger in width and quantity than that of the B520 sample (indicating more severe corrosion). As the aging temperature increases to 550 °C, the corrosion pits of 17-4PH transform into elliptical or circular shapes, with pit depths lower than those of the A480 and B520 samples. Further increasing the aging temperature leads to a trend of decreasing and then increasing pit density and size, with the smallest pit density and size observed at the 580 °C aging temperature.

Magnified images of the corrosion pits and surrounding corrosion products from Figure 10 are presented in Figure 11. Figure 12 displays the energy spectrum results corresponding to the red cross-marked points in Figure 11. As seen in Figure 11c,e,f, network-like corrosion traces surround the corrosion pits. These corrosion features form shapes resembling grain boundaries, with sizes ranging from 10 to 20 μm, similar to the original austenitic grains observed in the metallographic structure. Figure 11d–f shows that pitting is not confined to these grain boundary-like features only; rather, pitting marks are also distributed on the matrix. Moreover, as observed in Figure 11f, severe corrosion traces are more concentrated within the grains themselves, rather than at the grain boundaries.

The energy spectra of the matrix are shown in Figure 12(a1,b1,c1,d1,e1,f1), while the energy spectra of the corrosion products are represented in Figure 12(a2,b2,c2,d2,e2,f2). Table 3 shows the content of each element in Figure 12, where the sum of the content of the elements is not 100% because the content of Na and Cl on the surface of the sample has been removed. Comparing the matrix and corrosion product elements of all samples, it can be seen that the corrosion products of all samples are basically the same, mainly composed of oxides of Fe, Cr, Cu, and Nb. Research [11] suggests that in 17-4PH, Cr acts as a passivating element in corrosive environments, forming a dense metal oxide, such as Cr_2_O_3_, on the material surface, which provides protective benefits. However, with the progression of electrochemical testing, the passive film is broken, leading to pitting corrosion.

## 4. Discussion

The results from the electrochemical tests in Section 3.2 indicate that as the aging temperature increases, the corrosion resistance of 17-4PH first improves and then decreases, reaching its peak at 580 °C. The corrosion resistance of stainless steel is closely related to its microstructure and composition [17,36]. This study does not address the compositional changes in 17-4PH stainless steel; therefore, the factors influencing the corrosion resistance differences are sought within the material’s microstructure. Studies have shown that heat treatment alters the microstructure of steel, thereby affecting its corrosion resistance [17,18,19]. In Section 3.1, the microstructural changes of the samples, as observed through TEM, reveal that with increasing aging temperature, the number of dislocations in 17-4PH gradually decreases. Additionally, Cu elements tend to segregate, forming Cu-rich precipitates that precipitate from the matrix and gradually agglomerate and coarsen.

Studies on the relationship between dislocation density and the corrosion resistance of materials have shown that [26,27] high dislocation density can lead to an increase in corrosion-active sites within the material. However, when dislocations are evenly distributed, they can serve as nucleation sites for the passive film, providing a driving force for the formation of the passive film. This, in turn, contributes to the uniform formation and thickening of the passive film, thereby enhancing the material’s corrosion resistance [28,29]. Other studies [30,31] have indicated that excessive local variations in dislocation density can cause significant differences in electrochemical potential across different regions, which promotes metal dissolution and the formation of voids, leading to a reduction in the material’s corrosion resistance. In this study, Figure 4 and Figure 5 demonstrate that the dislocation distribution in 17-4PH is highly heterogeneous, with dislocations tending to concentrate within the laths or gather at the prior austenitic grain boundaries and lath boundaries. A480 exhibits the highest dislocation density, where the local dislocation density differences are so large that pitting corrosion preferentially occurs in these areas. As the electrochemical corrosion progresses, elongated deep pits are formed. As the aging temperature increases, the differences in dislocation density across different regions decrease, and the electrochemical potential difference between regions reduces. As a result, the deep pits formed in the B520 sample become narrower. When the aging temperature further increases to 550 °C, the dislocation density differences between regions decrease even further, and the pits become shallower, with their shape changing to either round or elliptical. The electrochemical corrosion differences between the C550, D580, E595, and F620 samples cannot be explained solely from the dislocation perspective; this must also be considered in the context of the copper element’s microstructure.

Copper, as an alloying element added to stainless steel, can enhance the corrosion resistance of the steel under certain environmental conditions [37,38,39,40]. The beneficial mechanism of copper is attributed to its deposition on the material surface, which inhibits the anodic dissolution process, a phenomenon that is particularly significant when the steel is immersed in corrosive media [38,41]. However, some researchers argue that in chloride-containing environments, copper may have a detrimental effect on the corrosion resistance of the material [14,33]. Other studies [41,42] suggest that the Cu-rich precipitates that segregate after aging exacerbate local electrochemical inhomogeneity, serving as preferential sites for pitting corrosion. As discussed earlier, the growth of Cu-rich precipitates is driven by the aggregation of fine copper-rich clusters [34,35]. In the present study, as the aging temperature increases, the Cu-rich precipitates first precipitate in large quantities from the matrix, and, as shown in Figure 5b, they are evenly dispersed within the matrix. Subsequently, these Cu-rich precipitates coarsen and agglomerate, as shown in Figure 5c, with their number significantly reduced compared to the previous stage. With further aging, as shown in Figure 5d, the Cu-rich precipitates continue to coarsen and agglomerate, and their number decreases to a certain extent. Although the copper-enriched phases are overall uniformly distributed, their density is significantly lower than that in Figure 5b, and some degree of segregation is observed within the matrix. This coarsening and agglomeration of Cu-rich precipitates, leading to a decrease in their density, results in a decline in the corrosion resistance of 17-4PH. As previously mentioned, the preferential sites for pitting corrosion, as shown in Figure 11d–f, are located within the grains, rather than at the grain boundaries, as shown in Figure 11c.

In summary, during aging of 17-4PH at 480 °C, copper elements only slightly agglomerate within the matrix, and no distinct Cu-rich precipitates are observed through TEM. Copper exhibits a uniform dispersion within the matrix, while dislocations are regionally distributed in large quantities. At this stage, dislocations play a major role in pitting corrosion, as the electrochemical potential difference between regions is too large, ultimately leading to the formation of elongated corrosion pits on the material’s surface. When the aging temperature increases to 520 °C, Cu-rich precipitate phases precipitate abundantly and uniformly within the matrix. The dislocation density decreases slightly compared to the 480 °C aging condition, but significant differences in dislocation density still exist between regions. The influence of dislocations continues to dominate the pitting corrosion, with the width of the elongated corrosion pits on the surface of the B520 sample being narrower than that on the A480 sample. As the aging temperature rises to 550 °C and beyond, the number of dislocations within 17-4PH further decreases, and the electrochemical potential difference between regions becomes smaller, leading to improved corrosion resistance. However, copper-enriched phases also undergo noticeable coarsening and agglomeration with increasing aging temperature, which causes segregation and results in a gradual deterioration of corrosion resistance. The preferential sites for pitting corrosion in 17-4PH are influenced by both dislocations and Cu-rich precipitates. At an aging temperature of 580 °C, while the dislocation density decreases to a certain extent, Cu-rich precipitates remain abundantly and uniformly distributed within the matrix. At this point, 17-4PH exhibits its best corrosion resistance. However, as the aging temperature continues to increase, the number of copper-enriched phases decreases and their size increases. The reduction in dislocation density differences is insufficient to compensate for the effects of Cu-rich precipitates segregation, which ultimately leads to a decline in the corrosion resistance of 17-4PH.

## 5. Conclusions

In this study, the microstructure of 17-4PH stainless steel subjected to different aging treatments was characterized, and the relationship between the corrosion resistance and microstructure of 17-4PH in a 3.5 wt% NaCl solution was investigated. The conclusions are as follows:As the aging temperature increases from 480 °C to 620 °C, the number of dislocations within the matrix gradually decreases. Precipitation behavior exhibited a sequential evolution: initial nucleation of Cu-rich precipitates at lower aging temperatures, followed by coarsening and agglomeration at elevated temperatures, accompanied by a corresponding decrease in precipitate population density.The corrosion resistance of 17-4PH stainless steel in 3.5 wt% NaCl solution first improved and then deteriorated with the aging temperature increasing, and the corrosion pit also changed from strip shape to oval shape. The corrosion resistance of the D580 sample was the best, and that of the A480 sample was the worst.Pitting susceptibility was governed by synergistic effects of dislocation distribution and copper-rich phase evolution. The regional distribution of dislocation leads to electrochemical potential difference between different regions, which leads to pitting corrosion. The degree of segregation of Cu-rich precipitates can also cause pitting corrosion.

## Figures and Tables

**Figure 1 materials-18-01823-f001:**
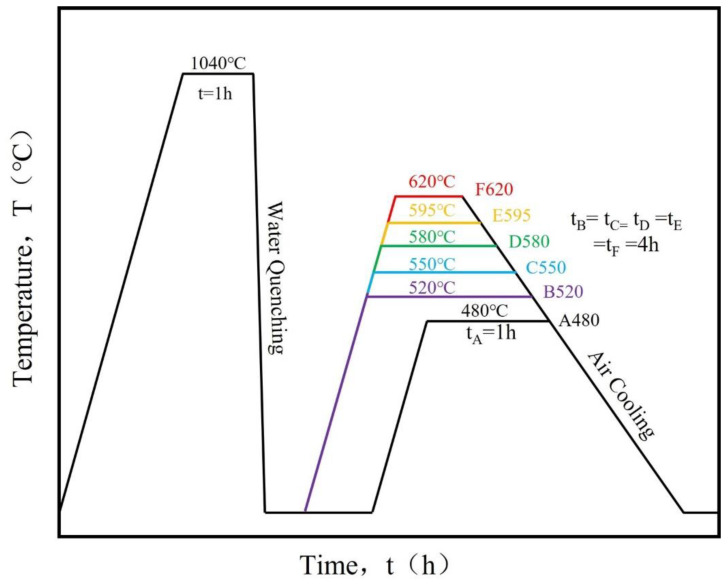
Heat Treatment Schematic for 17-4PH Stainless Steel.

**Figure 2 materials-18-01823-f002:**
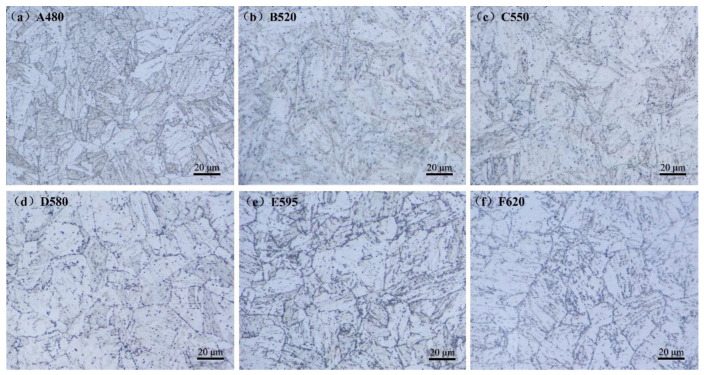
Light microscopy images of 17-4PH with different aging treatments: (**a**) A480; (**b**) B520; (**c**) C550; (**d**) D580; (**e**) E595; (**f**) F620.

**Figure 3 materials-18-01823-f003:**
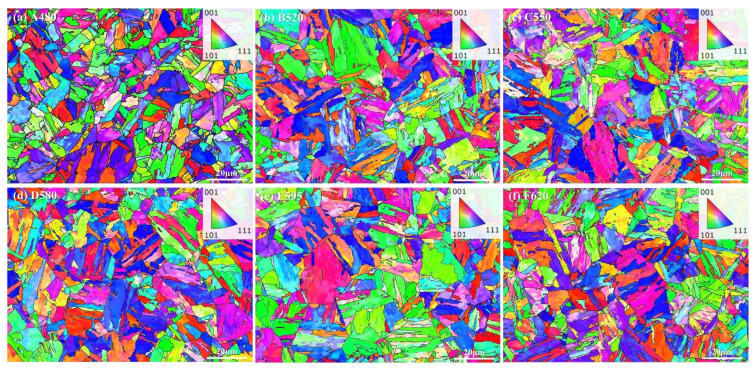
GB+IPF plots of 17-4PH with different aging treatments: (**a**) A480; (**b**) B520; (**c**) C550; (**d**) D580; (**e**) E595; (**f**) F620.

**Figure 4 materials-18-01823-f004:**
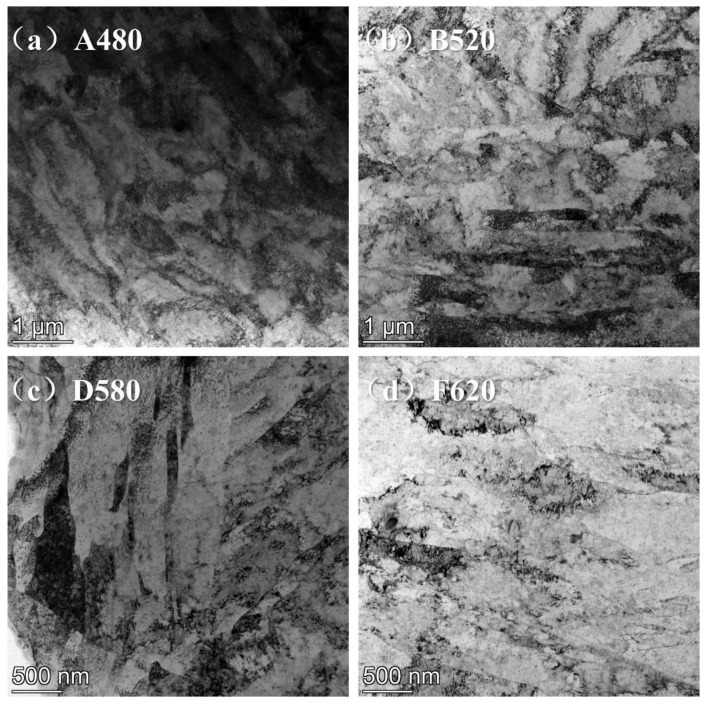
Dislocations of 17-4PH treated with different aging: (**a**) A480; (**b**) B520; (**c**) D580; (**d**) F620.

**Figure 5 materials-18-01823-f005:**
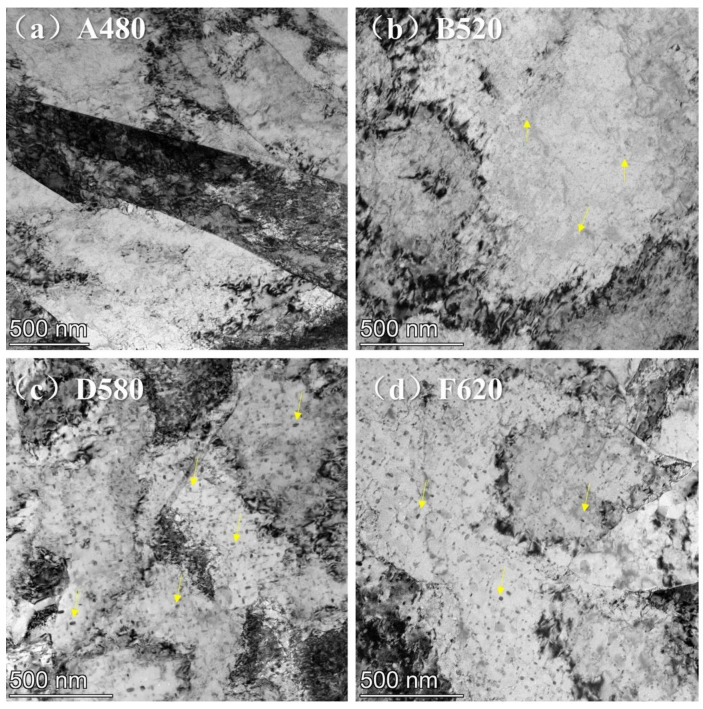
The precipitation of 17-4PH nanometer precipitates treated with different aging treatments: (**a**) A480; (**b**) B520; (**c**) D580; (**d**) F620. The yellow arrows indicate Cu-rich precipitates.

**Figure 6 materials-18-01823-f006:**
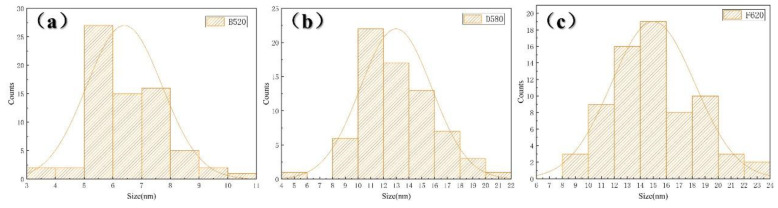
Size distributions of Cu-rich precipitates: (**a**) B520; (**b**) D580; (**c**) F620.

**Figure 7 materials-18-01823-f007:**
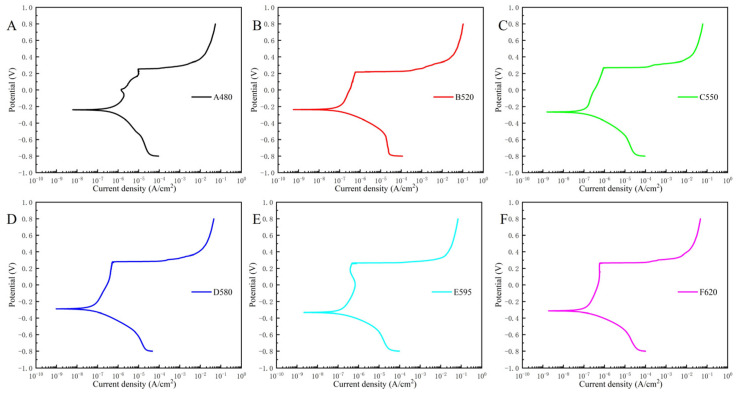
Potentiodynamic polarization curves of 17-4PH with different aging treatments in 3.5 wt.% NaCl solution: (**A**) A480; (**B**) B520; (**C**) C550; (**D**) D580; (**E**) E595; (**F**) F620.

**Figure 8 materials-18-01823-f008:**
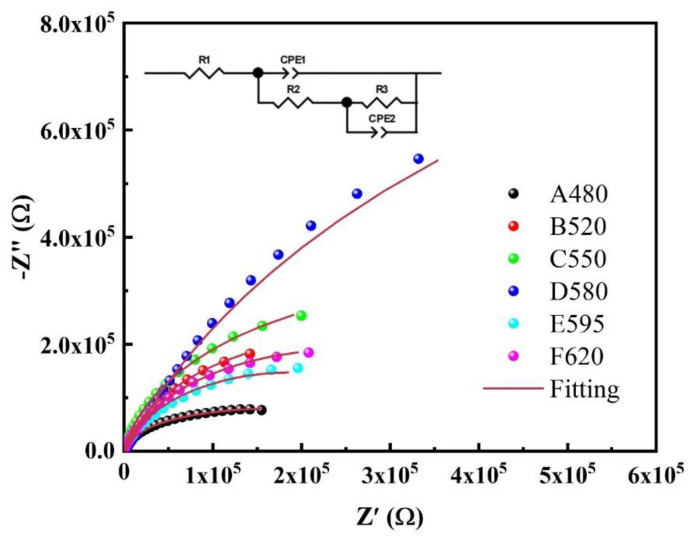
Nyquist plots and equivalent circuit diagrams of 17-4PH with different aging treatments.

**Figure 9 materials-18-01823-f009:**
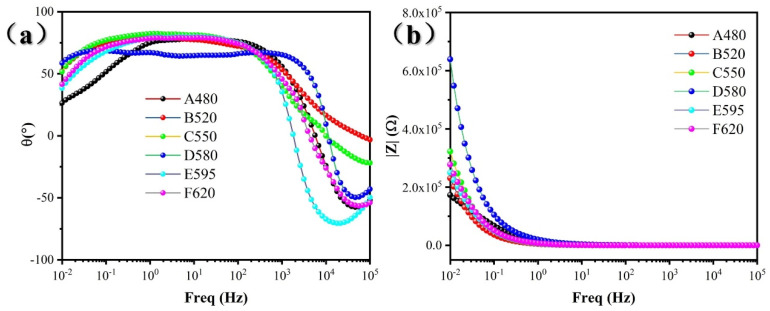
Bode diagrams of 17-4PH for different aging treatments: (**a**) Phase angle; (**b**) Impedance modulus.

**Figure 10 materials-18-01823-f010:**
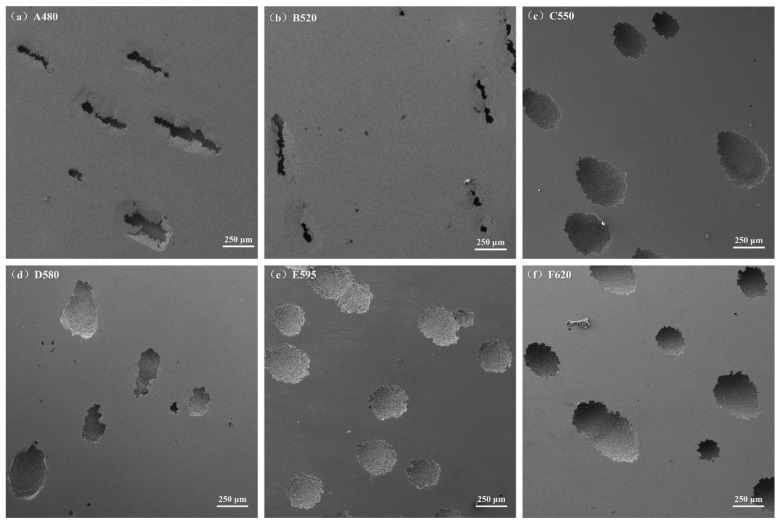
Surface morphology of the samples after electrochemical testing: (**a**) A480; (**b**) B520; (**c**) C550; (**d**) D580; (**e**) E595; (**f**) F620.

**Figure 11 materials-18-01823-f011:**
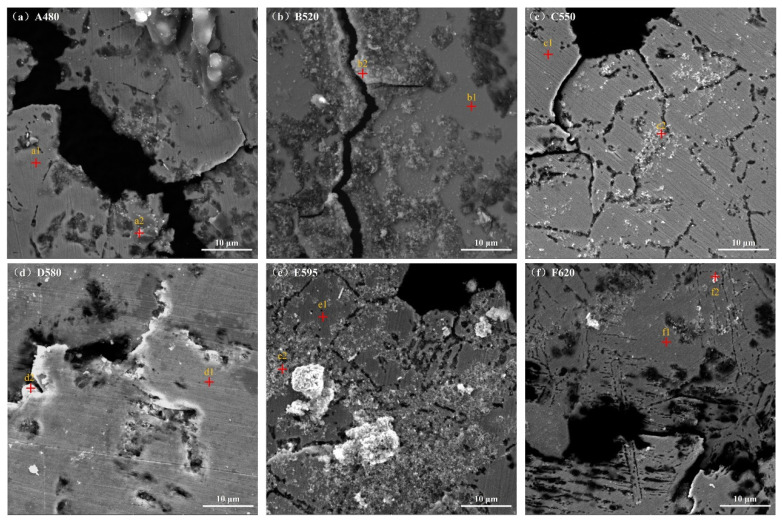
Corrosion pit and corrosion product morphology of the samples after electrochemical testing: (**a**) A480; (**b**) B520; (**c**) C550; (**d**) D580; (**e**) E595; (**f**) F620.

**Figure 12 materials-18-01823-f012:**
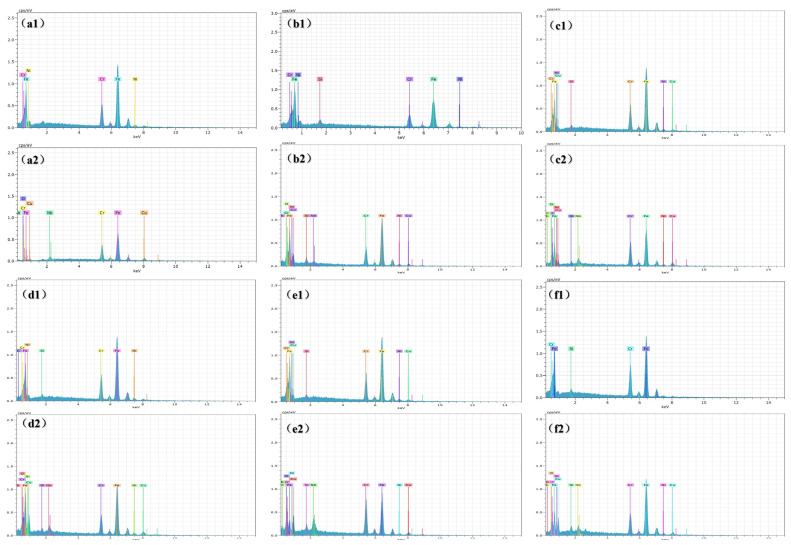
Energy spectrum of the substrate and corrosion products of the samples: (**a1**,**a2**) A480; (**b1**,**b2**) B520; (**c1**,**c2**) C550; (**d1**,**d2**) D580; (**e1**,**e2**) E595; (**f1**,**f2**) F620.

**Table 1 materials-18-01823-t001:** Chemical analysis (wt%) of the 17-4 PH stainless steel.

C	Si	Mn	Ni	P	S	Cr	Cu	Nb	Fe
0.031	0.53	0.58	3.87	0.018	0.0005	16.41	3.78	0.28	balance

**Table 2 materials-18-01823-t002:** Corrosion parameters of 17-4PH stainless steel with different aging treatments.

Samples	I_corr_ (nA/cm^2^)	E_corr_ (V_SCE_)	E_pit_ (V_SCE_)
A480	375	−0.241	0.254
B520	73.7	−0.236	0.217
C550	94.9	−0.265	0.266
D580	43.8	−0.288	0.279
E595	112	−0.327	0.265
F620	94.2	−0.311	0.262

**Table 3 materials-18-01823-t003:** Element content of each point in Figure 12.

	Fe (at%)	Cr (at%)	Ni (at%)	Cu (at%)	Nb (at%)	Si (at%)	O (at%)	C (at%)
a1	79.76	16.39	3.85					
a2	58.85	19.67		16.03	2.76		2.69	
b1	77.19	16.99	3.04			2.78		
b2	63.82	14.71	3.37	4.53	0.71	2.91	9.95	
c1	76.10	15.83	3.61	3.32		1.14		
c2	44.79	17.30	2.31	8.09	2.18	1.47	5.48	18.39
d1	65.03	14.09	3.59			0.87		16.42
d2	63.28	14.52	3.85	8.82	1.68	0.96	6.88	
e1	73.35	16.60	3.63	3.52		0.90		
e2	33.00	17.35	1.73	4.72	2.51	1.33	13.35	26.01
f1	79.33	19.62				1.05		
f2	55.00	12.14	2.31	5.44	1.15	1.33	7.62	14.32

## Data Availability

The datasets presented in this article are not readily available because the data are part of an ongoing study or due to technical.

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
