# Peer review of "The Effect of Aging Treatment on the Corrosion Behavior of 17-4PH Stainless Steel"

_materials, 2025, doi:10.3390/ma18081823_

Round 1

Reviewer 1 Report

Comments and Suggestions for Authors

Dear Authors,

Thank you for the Chance to review the paper entitled “The Effect of Aging Treatment on the Corrosion Behavior of 17-4PH Stainless Steel”. Below you will find a few remarks regarding your work.

Table 1 – Was the chemical composition analyzed by the authors or was it provided by the manufacturer? If it was analyzed by the authors then the method of analysis and accuracy should be mentioned in the manuscript.

Why did the authors chose water quenching instead of e.g. oil? Authors could also discuss cryogenic quenching in liquid nitrogen in their future works as there are numerous papers on the beneficial effects of quenching in -200 °C. This is something to think about in the future.

Why A480 was only treated for 1 hour, and not 4 hours as the other (higher) temperatures? Was there a specific reason for shortening the time here?

Was heat treatment (both at 1040 °C and at aging temperatures) done with some sort of protective atmosphere?

Overall, the paper shows good merit, it was well-thought and the results are clearly presented and discussed. It was a pleasure to read your paper.

Reviewer 2 Report

Comments and Suggestions for Authors

Comments are provided in the attached PDF file.

Round 2

Reviewer 2 Report

Comments and Suggestions for Authors

Although the authors have made some corrections to the text, they have neither addressed nor justified most of the comments. This is particularly evident in regard to the quality of the results presented in Figures 12 and 13. Additionally, the results presented in Figure 12 should be supplemented with tables for a more comprehensive representation.
